# Bubble Formation in Pulsed Electric Field Technology May Pose Limitations

**DOI:** 10.3390/mi13081234

**Published:** 2022-07-31

**Authors:** Isaac Aaron Rodriguez Osuna, Pablo Cobelli, Nahuel Olaiz

**Affiliations:** 1Laboratorio de Sistemas Complejos, Departamento de Computación, Instituto de Física del Plasma, Facultad de Ciencias Exactas y Naturales, Universidad de Buenos Aires, Buenos Aires 1428, Argentina; isaacorp94@gmail.com; 2Departamento de Física, Facultad de Ciencias Exactas y Naturales, Universidad de Buenos Aires, Buenos Aires 1428, Argentina; nahuelolaiz@gmail.com; 3Instituto de Física de Buenos Aires, CONICET, Ciudad Universitaria, Buenos Aires 1428, Argentina

**Keywords:** electroporation protocols, electric field, current density, pulse electric field, electrode coverage

## Abstract

Currently, increasing amounts of pulsed electric fields (PEF) are employed to improve a person’s life quality. This technology is based on the application of the shortest high voltage electrical pulse, which generates an increment over the cell membrane permeability. When applying these pulses, an unwanted effect is electrolysis, which could alter the treatment. This work focused on the study of the local variations of the electric field and current density around the bubbles formed by the electrolysis of water by PEF technology and how these variations alter the electroporation protocol. The assays, in the present work, were carried out at 2 KV/cm, 1.2 KV/cm and 0.6 KV/cm in water, adjusting the conductivity with NaCl at 2365 μs/cm with a single pulse of 800 μs. The measurements of the bubble diameter variations due to electrolysis as a function of time allowed us to develop an experimental model of the behavior of the bubble diameter vs. time, which was used for simulation purposes. In the in silico model, we calculated that the electric field and observed an increment of current density around the bubble can be up to four times the base value due to the edge effect around it, while the thermal effects were undesirable due to the short duration of the pulses (variations of ±0.1 °C are undesirable). This research revealed that the rise of electric current is not just because of the shift in electrical conductivity due to chemical and thermal effects, but also varies with the bubble coverage over the electrode surface and variations in the local electric field by edge effect. All these variations can conduce to unwanted limitations over PEF treatment. In the future, we recommend tests on the variation of local current conductivity and electric fields.

## 1. Introduction

The ability to enhance absorption [1], extraction of molecules [2,3], and cell inactivation by non-thermal processes and cellular manipulation techniques with a pulsed electric field has long been discussed [4]. The phenomena of absorption, extraction, and cell inactivation are directly related to a different mechanism of permeation and surface modification of the surface of cell membrane [5,6,7,8]. During the last decade, PEF demonstrated that it can alter the permeability of the membrane without increasing temperature and undesirable effects [9,10]. This PEF was explored in different scientific domains such as medicine [6], biotechnology [11], and environmental preservation [4]. Application of PEF therapy is a procedure using an intense but short electric pulse that provokes either permanent permeabilization of cancer cells or destabilizes the cell membranes and intracellular components to which the cells are unable to repair resulting in their death [12]. The application of PEF processing has been studied in food products and has been studied for several decades as a useful method for the pasteurization of food. It is also applied in improving the extraction of sugars and other cellular contents [13].

PEF treatment generally occurs in an aqueous medium [4], often generating bubbles [14] (due to electrolysis of water) or increasing their volume (due to dielectric breakdown of the gas bubbles that generate a spark that can volatilize the liquid and produce more vapor) [15]. Some studies show standard electroporation treatments of eight pulses and 100 μs [7] and its dose can be simplified in terms of energy. It can be summarized in a pulse of 800 μs [16], and for this reason, the phenomenon of bubble formation was studied in the presence of a high electric field for a pulse of 800 μs. This bubble formation happens according to the following reaction set.

The overall reaction between anode and cathode during the PEF application is:(1)$$2H_{2}O_{\left(l\right)}<=>H_{2(g)}+O_{2(g)}$$

Overall cathodic reaction for hydrogen evolution
(2)$$2H_{2}O_{\left(l\right)}+2e^{-}<=>2OH_{\left(g\right)}^{-}+H_{2(g)}$$

Overall anodic reaction for oxygen evolution
(3)$$4OH_{\left(l\right)}^{-}-4e^{-}<=>2H_{2}O_{\left(l\right)}+O_{2(g)}$$

A study of the variations of current density change as function of the bubble diameter was carried out [17] demonstrating an exponential increment of the current density with the electrode coverage by a gas film. The variation of the local current density can affect some electroporation treatments based on the amount of applied energy, like a inactivation of bacteria, antibiotics and another cellular organisms [18,19,20], extraction of bio-components [21,22,23], gen therapy [24,25], microfluidics chamber designs [26,27], food industry processes [28,29], hydrogen production [30] and another process based of the mass transfer area [31,32]. Another undesirable effect of the current density changes is the increase of temperature and dielectric breakdown [33,34] as a function of electrical conductivity [35]. Specifically, the formation of bubbles is starting to gain importance in the field of hydrogen production as fuel through water electrolysis, and study of how the useful surface of the electrode decreases and the concentration of reagents on the surface is increasing [11].

Understanding the fundamental mechanisms governing gas bubble evolution and gas bubble interaction in fluids will have remarkable implications in the fields of biomedicine and biotechnology. This work shows the first results of the gas bubble coverage interaction dynamics during the application of PEF and their effects on applied electric field distribution.

## 2. Materials and Methods

In vitro modeling is based on the application of the PEF different in a microwell. Figure 1 shows a scheme of the experimental setup for bubble tracking. Microwell electropermeabilization is a plastic well with a 6 mm diameter and is 10 mm high with two electrodes (parallel solid steel 316 needles of 0.45 mm diameter and 10 mm long, separated 0.5 mm from each other). Its microwell was filled with 0.16 mol/dm of NaCl, with a conductivity of 2365 μS/cm [36]. Different dosages (30 V, 60 V, and 100 V; a pulsed of 800 μs, generated with Micropulse generator, Einsted S.A, Buenos Aires City, BA AR) of a PEF were measured with a digital oscilloscope (InfiniiVision DSO-X 2012A-SGM, Agilent Technologies, Santa Clara, CA, USA). During the PEF delivery, video capturing of the image area was performed by a 1000CASIO EX-Fh25 high-speed camera incorporated in an Olympus BX41 microscope (Tokyo, Japan). The video was acquired at 10 kfps with a 1280 × 720 pixel resolution. Visual front tracking of the bubble emerging from the cathode was obtained by the image process. Illumination was achieved by a high-intensity white LED. Video images were processed and analyzed with the ImageJ graphic package [37]. The assay was performed in the vertical position of electrodes, with respect to the edge of the electrode.

### 2.1. Numeric Simulation of Electric Field Distribution

To estimate the increase in the electric field due to the edge effect around the bubble, the finite element method was used in COMSOL 4.3 applying the following set of equations that describe the phenomenon of direct current, which adjusts to the width pulse:(4)$$\Delta J=Q_{i,j}$$
(5)$$J=\sigma E+\frac{\delta D}{\delta t}+J$$
(6)E=ΔV

The simulation we use a core I7 9700K, with 40 GB of memory ram. The mesh configuration for general calculation are shown in Table 1:

The general simulation domain is shown in Figure 2. For the study of the dynamic changes of current density and electric field around the bubble, we selected a 0.1 × 0.1 mm Figure 3 area between the two electrodes and studies these variations as a function of the bubble diameter; in turn, the bubble diameter is a function of time.

### 2.2. Electrode Insulation by Gas Film Generation Model

The gas film coverage percent can be estimate by the Vogt Equation [38]
(7)$$\Theta \left(t\right)=0.604\frac{V_{g}}{A}*\frac{t_{r}}{V_{b}\left(t\right)}$$
where Θ: the fraction of coverage by gas phase; $V_{g}$: volume rate of gas; *A*: the electrode surface area; $t_{r}$: treatment time; $V_{b}$: bubble volume.

The max current density can be calculated by the Vogt model as a maximum of the function max (Θ(I/A)). From this model, we can know the possible max density current. With this information and knowing the current increase rate (measurement with the oscilloscope), we can estimate the time that the system achieved the max current value, with the next equation:(8)$$\Theta =\frac{I_{max}-I_{i}}{m}$$
where $I_{max}$ is the maximum current density; $I_{i}$ is the initial current at *t* = 0.01 μs; *m* is rate of change of the current density.

### 2.3. Bubble Diameter Measurement

The experimental measurement was carried out by taking the measurement of the bubbles located at 0°, 90° and 180° of the cathode in order to obtain the different variations of the bubble diameters (see Figure 4 for the bubble diameter distribution around the electrode and see Figure 5 for experimental pictures).

Example of the measurement of bubble diameter:

## 3. Results and Discussion

Here we will present the results of the overall simulation and local simulation around the bubble. First is shown the variations of the electric field by edge effect around the bubble, and second the variations of the local current density by these local variations of the electric field.

When using 30 V and 60 V, we did not appreciate significant changes in the current. However, when using 100 V, we observed (Figure 6) an increase in the current rate change over time. These changes in current are due to the electrode insulation by gas film formation. The current change rate is shown in Table 2.

On another side, the bubble diameter in the function of time presents a linear tendency. This corresponds with the overall reactions [14] because the bubble diameter increase with the accumulated applied energy [39]. We observed results according to the distribution energy estimations (Figure 7D). The larger diameter is present at 0° and decreases as the radiating distribution increases in relation to the opposite electrode (Figure 7A–C).

The results of the simulation of the bubble domain (Figure 3) as shown below. First, we analyzed the variation of the electric field around the bubble to estimate the variations of this by edge effect around the bubble. In Figure 8A, we can observe an exponential increase of the electric field with the variations of the bubble diameter or the electrode coverage area because when the bubble increases in diameter, the edge effect of the electric field increases (Figure 8B). On the other hand, the general electric field between the two electrodes is normal, around 2 KV/cm for all bubbles diameter (Figure 8C).

Knowing that the electric field varies with bubble diameter, the next step was to study the variations of current density as a function of the bubble. The simulation shows that the variation of this variable has the same tendency are the increase in the electric field as a function of the diameter of the bubble [17] (Figure 8A) and the current density concerted in the edge of the bubble and the electrode (Figure 8B), setting a lower value at the pole of the sphere.

Analyzing the results of the simulations, we can relate the increase in the current density. The results follow the tendency of Equation (Equation 7) [38] (Figure 9). We do not include the 30 V assay in Figure 9, because the currents change are too small to appreciate changes in the range of study. For the 60 V and 100 V, the max peaks of current are shown in Table 3.

## 4. Conclusions

According to the model for hydrogen formation by the electrolysis process, [14,39] the experimental measurements in Figure 1 are in accordance with the theoretical model. The bubble diameter presents a lineal tendency (Figure 7) with the time because the hydrogen formation is lineal with the quantity of applied energy. The energy distribution shows that the highest value of energy is °0; for this reason, the bubbles in this region are bigger than other regions (90–180°).

We can observe current variations along the pulse for the 100 V assay (Figure 6), but for the case of 30 V and 60 V we have not observed significant variations. This is because the electrode coverage for these assays was negligible. The simulations around the growing bubble show an increment of the local electric field in the cathode surface (Figure 8, because of this the current density suffers an increase with an exponential tendency (Figure 8). From these simulations, we can estimate the local current density changes as a function of the electrode coverage (Figure 9) and predict the peak of current density, critical electrode coverage and the time condition that can be achieved for the determinate electric field, electrical conductivity and applied voltage (Table 3).

The local variations of current density affect some electroporation protocols based on the amount of energy transfer [26,27,40], affecting the local electric field distribution and the final result of the treatment. This work studied these variations according to to obtain more precise results in the simulations and in order to avoid the electrode insulation by the formation of the gas film. Another interesting point is that this variation of an electric field induces an electrical discharge (spark) and, in general, this is an undesirable phenomenon, especially in the area of IRE (irreversible electroporation) applied to tumor ablations [41].

The geometry of the electrodes is very important to reduce the number of bubbles trapped on the surface of the electrodes in order to reduce the increase in current density. In the case of hydrogen production, the increase in current density can lead to a significant reduction in the efficiency of the process and increase the cost of hydrogen production per kilogram, making the bubble formation process of wide interest study for this field [11].

## Figures and Tables

**Figure 1 micromachines-13-01234-f001:**
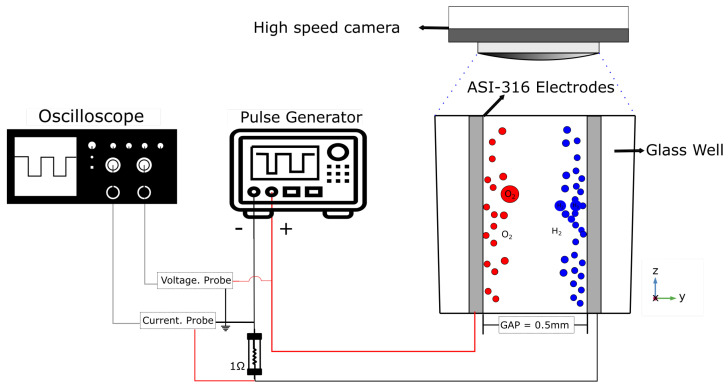
Microwell electropermeabilization setup.

**Figure 2 micromachines-13-01234-f002:**
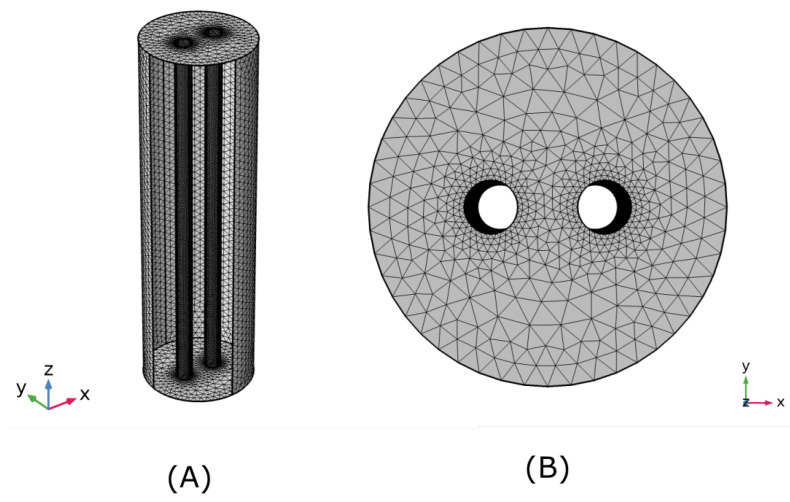
Configuration of the mesh for finite element simulation.(**A**) Mesh ISO-View. (**B**) Mesh Top-view.

**Figure 3 micromachines-13-01234-f003:**
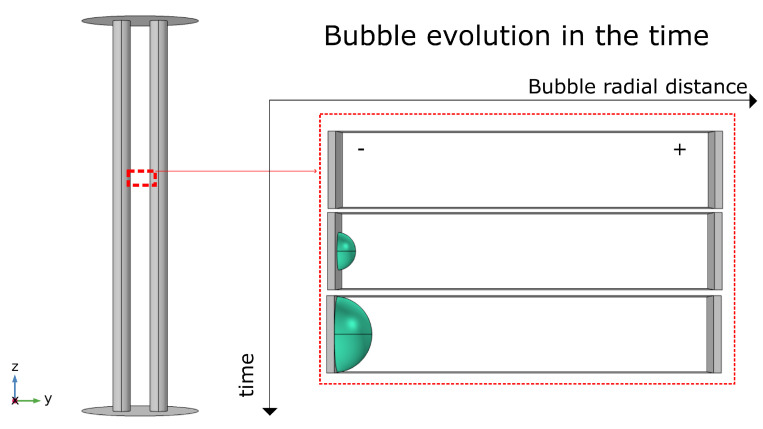
Bubble Study domain, an example of bubble diameter at different instants of time.

**Figure 4 micromachines-13-01234-f004:**
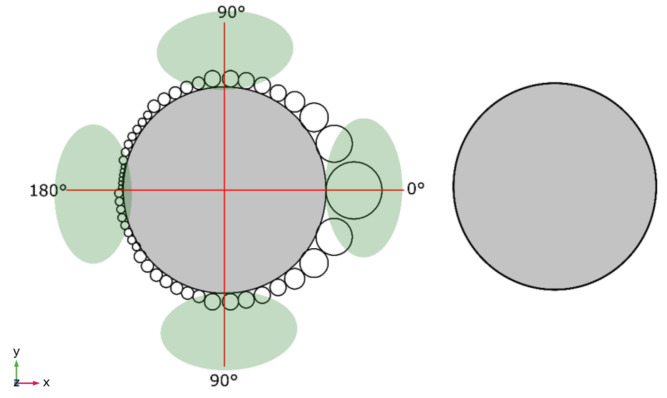
Description of bubble measurement technique.

**Figure 5 micromachines-13-01234-f005:**
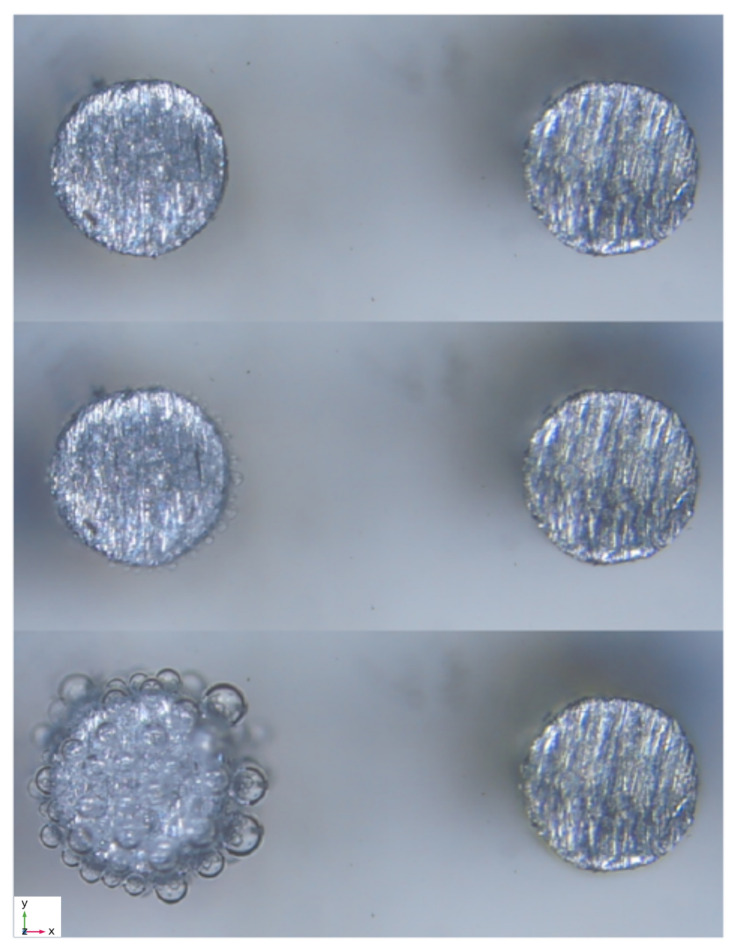
Bubble measurement photo-grams examples for 100 V assay.

**Figure 6 micromachines-13-01234-f006:**
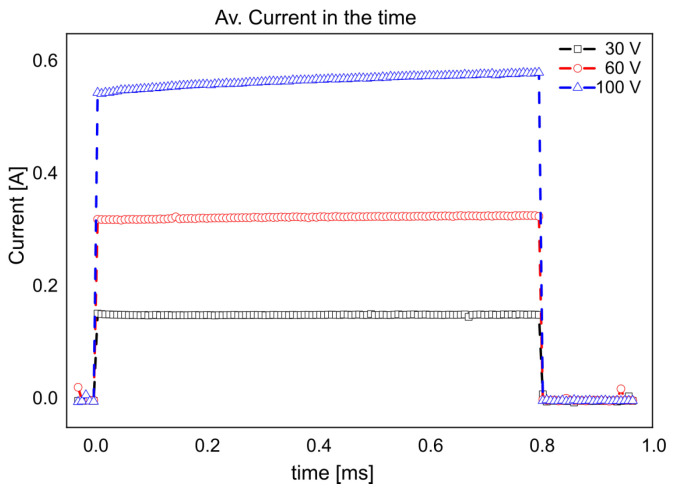
Current measurements over time.

**Figure 7 micromachines-13-01234-f007:**
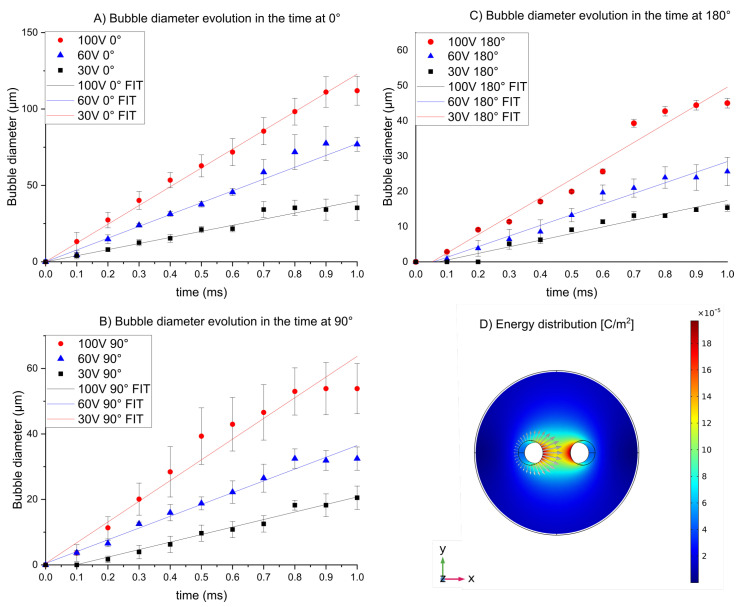
Average Bubble diameter as function of time and simulations of energy distribution and current density vectors. (**A**) Bubble measurements at 0°. (**B**) Bubble measurements at 90°. (**C**) Bubble measurements at 180°. (**D**) Energy distribution and current density vectors. (Data set in Appendix A).

**Figure 8 micromachines-13-01234-f008:**
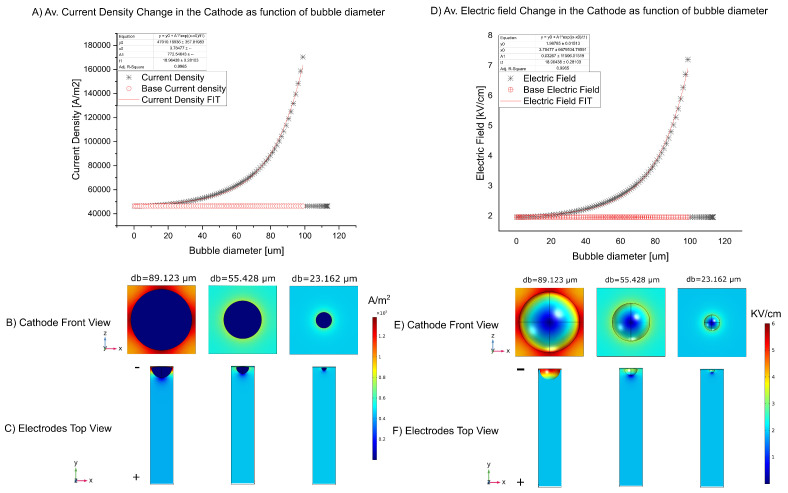
(**A**) Average Current density changes as function of bubble diameter for 100 V Assay at 0°. (**B**) Current density change as function of the bubble diameter in the cathode surface. (**C**) Current density distribution between the electrodes. (**D**) Electric field changes as function of bubble diameter for 100 V Assay 0°. (**E**) Electric field change as function of the bubble diameter in the cathode surface. (**F**) Electric field distribution between the electrodes.

**Figure 9 micromachines-13-01234-f009:**
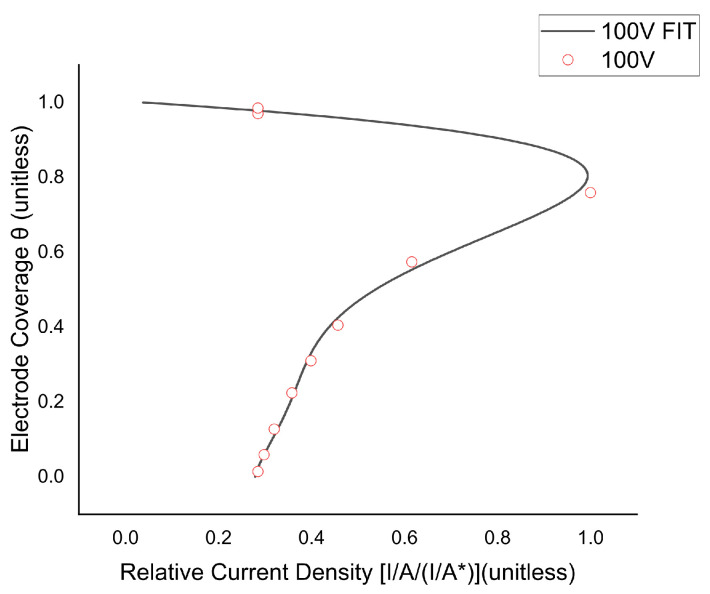
Theoretical and simulated variations of current density in the surface of the electrode for 100 V treatment (2 KV/cm Electric field). I/A = Actual current density; I/A* = Maximum current density; Θ = Electrode coverage fraction.

**Table 1 micromachines-13-01234-t001:** Mesh Configuration parameters.

Mesh Configuration [Units]	Value
Max. Element Size [mm]	0.200
Min. Element Size [mm]	0.002
Element Growing Rate [unitnlessl]	1.300
Curvature factor [unitless]	0.200
Resolution of narrow regions [unitless]	1.000

**Table 2 micromachines-13-01234-t002:** Current changes over time.

Voltage [V]	Current Change [A/ms]
30	0.00070
60	0.00893
100	0.04118

**Table 3 micromachines-13-01234-t003:** Max current density and peak rise time for 100 V.

Voltage [V]	Θ at Max Current	Current Peak [A/m${}^{2}$]	Peak Rise Time [ms]
100	0.76	165,139	43.078

## Data Availability

Not applicable.

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
