# Peer review of "Bubble Formation in Pulsed Electric Field Technology May Pose Limitations"

_micromachines, 2022, doi:10.3390/mi13081234_

Round 1

Author Response

Reviewer 1, Round 1.

Hello dear, thanks for your review. To continuation, I insert your revisions and my corrections.

The paper titlel “Forming bubble in Pulsed Electric Fields Technology may impede the treatment” represent an interesting study, but in my opinion, before it is considered for publication, some changes should be made.

In the text are present more and more mistakes, for example:

Page 1, line 13. this study demonstrated that the electric….. Capital letter  OK

Page 1, line 13. of the local electric field.It is also important… Add space OK

Page 1, line 17. electrodes.In the future …. Add space OK

Page 2, line 40. An study of the variations…. Correct the mistake  OK

Page 2, line 40. ,Extraction….. Add space and replace comma with dot. In this case it is not necessary because it is a list of topics separated by commas

Page 2, line 44 to 52. Check the period, punctuation is not clear, Same that anterior review.

Pag.2, line 57, Microwell electrpopermeabilization …. Correct the mistake OK

……

The amount and quality of the manuscript are poor, the authors should increase the introduction and the discussion of the results.

We add more information about the bubble formation for the production of hydrogen as fuel and how affects the Bubble formation that process. We have enough evidence to show how bubble formation affects the electrolytic treatment of water, reducing its efficiency.

Reviewer 2 Report

This manuscript offers a concise yet compelling report on how forming bubble(s) in pulsed electric fields (PEF) subjected media can affect this technology and its applications. A detailed experimental scheme for bubble tracking is used. The chosen modelling efforts are realized by employing the finite element method as implemented in the COMSOL code, thus ensuring results which are well described. The nice comparative experimental-to-modelling context of results throughout the study strongly contribute to the elaboration of a successful and reliable protocol for scrutinization of bubbles in PEF – a phenomenology which is not much studied yet. From practical point of view, the reported results thus bring new knowledge and certainly represent an original contribution in the present context.

The authors chose an adequate structure of the manuscript – an excellent point of departure for such a study. Finally, the authors provided a balanced realistic and nicely illustrated presentation of their results and corresponding analysis that is of much scientific and practical interest and adds new knowledge to the field.

In my opinion, the fine detailing in the present work, the insightful and balanced discussion of the results, as well as the very good figures, permit competent readers to utilize the manuscript as a guidance for future work. Consequently, this manuscript presents an efficient and beneficial basis for promoting and solving next step challenges in this field.

Moreover, the manuscript benefits from a clear motivation and it is an easy and informative read. The manuscript is also excellent in terms of clarity and accuracy of language.

The present manuscript is a significant contribution, this work once published would be quite useful as well as instructive and suggestive in terms of further studies and to a wider readership.

There are some minor issues with this already excellent manuscript that will need to be addressed before becoming suitable for publication, i.e., it can be considered for publication after a minor revision:

1: I’m not so sure about the full adequacy and attractiveness of the tittle. It would be better to avoid rationalizing it to something like: “Babble formation in Pulsed Electric Field Technology may pose limitations to some of its most important applications in medicine”

2: In the introduction, the authors partly miss aspects of the general scope of modeling of different media and material systems in electric fields such tasks may be even approached by ab-initio simulations. Examples include Thin Solid Films 517 (2008) 1106-1110; The Journal of Physical Chemistry C 121 (2017) 26125-26132. Such works evidence the broadness of modelling options.

3: What will be the range of thermal stability of the suggested set-ups for bubble formation? Authors mentioned that the thermal effects were despised due to duration of pulses and that’s very right, but it is conceptually important to expose to what extent the general thermal range impacts the presented results.

4: Usually, it is considered unnecessary to include citations/references in the conclusions.

5: Spell-check and stylistic revision of the paper are still necessary. Some long sentences, misspellings, etc., still are noticeable throughout the text.

Author Response

Hello dear, thanks for your review. To continue, I insert your revisions and my corrections.

This manuscript offers a concise yet compelling report on how forming bubble(s) in pulsed electric fields (PEF) subjected media can affect this technology and its applications. A detailed experimental scheme for bubble tracking is used. The chosen modeling efforts are realized by employing the finite element method as implemented in the COMSOL code, thus ensuring results that are well described. The nice comparative experimental-to-modeling context of results throughout the study strongly contributes to the elaboration of a successful and reliable protocol for scrutinization of bubbles in PEF – a phenomenology that is not much studied yet. From a practical point of view, the reported results thus bring new knowledge and certainly represent an original contribution in the present context.

The authors chose an adequate structure of the manuscript – an excellent point of departure for such a study. Finally, the authors provided a balanced realistic, and nicely illustrated presentation of their results and corresponding analysis that is of much scientific and practical interest and adds new knowledge to the field.

In my opinion, the fine detailing in the present work, the insightful and balanced discussion of the results, as well as the very good figures, permit competent readers to utilize the manuscript as a guidance for future work. Consequently, this manuscript presents an efficient and beneficial basis for promoting and solving next step challenges in this field.

Moreover, the manuscript benefits from a clear motivation and it is an easy and informative read. The manuscript is also excellent in terms of clarity and accuracy of language.

The present manuscript is a significant contribution, this work once published would be quite useful as well as instructive and suggestive in terms of further studies and to a wider readership.

There are some minor issues with this already excellent manuscript that will need to be addressed before becoming suitable for publication, i.e., it can be considered for publication after a minor revision:

1: I’m not so sure about the full adequacy and attractiveness of the tittle. It would be better to avoid rationalizing it to something like: “Babble formation in Pulsed Electric Field Technology may pose limitations to some of its most important applications in medicine”

Thanks for the tittle we considered this tittle for the publications. But we considered that the results of the publication can be extensible to hydrogen productions and medicine, due the changes of current density. So the finally title can be: Babble formation in Pulsed Electric Field Technology may pose limitations to some of its most important applications in medicine and hydrogen productions.”

2: In the introduction, the authors partly miss aspects of the general scope of modeling of different media and material systems in electric fields such tasks may be even approached by ab-initio simulations. Examples include Thin Solid Films 517 (2008) 1106-1110; The Journal of Physical Chemistry C 121 (2017) 26125-26132. Such works evidence the broadness of modelling options.

3: What will be the range of thermal stability of the suggested set-ups for bubble formation? Authors mentioned that the thermal effects were despised due to duration of pulses and that’s very right, but it is conceptually important to expose to what extent the general thermal range impacts the presented results.

For one pulse of 800us in ultra pure water, the total current amount suggests variations of +-0.1°C  for this motive we are despicable the thermal variation. We include this variations range In the publication.

4: Usually, it is considered unnecessary to include citations/references in the conclusions.
Yes is correctly, but in our case we decide to include for show the similarity of the results of other works in DC voltages with our PEF work.

5: Spell-check and stylistic revision of the paper are still necessary. Some long sentences, misspellings, etc., still are noticeable throughout the text.

We did the spell checking.

Reviewer 3 Report

This paper studies the formation of bubbles during pulsed electric fields use and the resulting limits in electroporation applications. The subject is interesting, but several questions arise from the reading of the paper.

1) Authors have to give a general revision in the English of the article, which makes it very difficult to read and understand. The paper is full of incompressible sentences,

for example:

Line 142: In this work studied these variations according to obtain more precise results in the simulations and in order to avoid the electrode insulation by the formation of the gas film. 

Line 134: because of this the current density suffers an increase with an exponential tendency (Figure ??).

Line 133: electrode coverage for theses assays was despicable

Line 132: we not observed significantly variations.

Line 128: because the hydrogen formation is lineal with quantity of applied energy

Line 119: Analyzing the results of the simulations we can relationship the increase of the current 119 density. 

Line 87: For the experiences of 30V and 60V , we no appreciated a significant changes of current. however for 100V experiences we can observed (Figure 6) an increase of the current rate change in the time. these changes of current are due the electrode insulation by gas film. The current change rate are show in the table 2

2) What is the effect of the temperature in the effect described,

3) In the Introduction section authors talk about the electroporation effect and PEF use in biomedical, food and environment applications. In these applications PEF protocols are use from 100s ns to 100s µs range, from Hz to MHz operation, with voltage amplitudes up to 10s of kV, currents up to 100s of A, using monopolar to bipolar pulses. So, the protocol used, 800 µs pulse and 2 kV/cm, applied 100 V in 0.5 mm, how is this representative of the PEF applications described.

4) Some figures are very difficult to understand:

5) Authors reference past work related to PEF and electroporation applications, and also work relative to electrolysis phenomena. However, authors seemed have missing some important references regarding the subject of bubble formation due to PEF application:

Samo Mahnič-Kalamiza, Damijan Miklavčič, Scratching the electrode surface: Insights into a high-voltage pulsed-field application from in vitro & in silico studies in different fluid, Electrochimica Acta, Volume 363, 2020, 137187, ISSN 0013-4686, https://doi.org/10.1016/j.electacta.2020.137187.

Author Response

Hello dear, thanks for your review. To continuation, I insert your revisions and my corrections

This paper studies the formation of bubbles during pulsed electric fields use and the resulting limits in electroporation applications. The subject is interesting, but several questions arise from the reading of the paper.

1) Authors have to give a general revision in the English of the article, which makes it very difficult to read and understand. The paper is full of incompressible sentences,

for example:

Line 142: In this work studied these variations according to obtain more precise results in the simulations and in order to avoid the electrode insulation by the formation of the gas film.

Line 134: because of this the current density suffers an increase with an exponential tendency (Figure ??).

Line 133: electrode coverage for these assays was despicable

Line 132: we did not observe significantly variations.

Line 128: because the hydrogen formation is linear with quantity of applied energy

Line 119: Analyzing the results of the simulations we can relate the increase of the current 119 density.

Line 87: For the experiences of 30V and 60V , we did not appreciate a significant changes of current. however for 100V experiences we can observed (Figure 6) an increase of the current rate change in the time. these changes of current are due to the electrode insulation by gas film. The current change rate are shown in the table 2

Ok thanks for the observations, In this time we used the spell-checking and grammar

2) What is the effect of the temperature in the effect described.

We include a range of temperature variation now. But the effect is universal knower is by ideal gas law and affects the bubble volume as V=NRT/P and the electrical conductivity.

3) In the Introduction section authors talk about the electroporation effect and PEF use in biomedical, food and environment applications. In these applications PEF protocols are used from 100s ns to 100s µs range, from Hz to MHz operation, with voltage amplitudes up to 10s of kV, currents up to 100s of A, using monopolar to bipolar pulses. So, the protocol used, 800 µs pulse and 2 kV/cm, applied 100 V in 0.5 mm, how is this representative of the PEF applications described.

Yes is correctly, but the pulse durations into the medical applications can be change according the Electric Field, Pulse number, Tissue geometry and physical properties. We do not want to represent with exactitude the medical aplications of PEF in this paper. If we want to show how the electrodes are insulated with the gas layer that can be generated and this same henomenon can affect the treatments depending on the amount of energy applied.

 4) Some figures are very difficult to understand:

Which are?

 5) Authors reference past work related to PEF and electroporation applications, and also work relative to electrolysis phenomena. However, authors seemed to have missing some important references regarding the subject of bubble formation due to PEF application:

Samo Mahnič-Kalamiza, Damijan Miklavčič, Scratching the electrode surface: Insights into a high-voltage pulsed-field application from in vitro & in silico studies in different fluid, Electrochimica Acta, Volume 363, 2020, 137187, ISSN 0013-4686, https ://doi.org/10.1016/j.electacta.2020.137187.

 OK

Round 2

Reviewer 1 Report

Authors have fixed some of the text, but there continue to be several errors, so I advise them to pay more attention to it.

1)      I advise authors to check the title:

 : Bubble formation in Pulsed Electric Field Technology may pose limitations to some of its most important applications in medicine and hydrogen production;

2)      pag. 5, line 120 and 121, check the error

(Figure ?? (A)) the current density s concerted at the edge of the bubble and the electrode (Figure ?? (B)) setting…

3)      pag. 7, line 139. check the error: an exponential tendency (Figure ??).

4)      Check punctuation, missing points. 

Author Response

Hello dear, thanks again for your review. To continue I copy your comments in this text and answer you line by line.

1)      I advise authors to check the title:

 : Bubble formation in Pulsed Electric Field Technology may pose limitations to some of its most important applications in medicine and hydrogen production;

R- We have made this title change at the recommendation of one of the reviewers. We believe that it can have a greater impact than the previous one.

2)      pag. 5, line 120 and 121, check the error - Fixed

(Figure ?? (A)) the current density s concerted at the edge of the bubble and the electrode (Figure ?? (B)) setting…

3)      pag. 7, line 139. check the error: an exponential tendency (Figure ??).-Fixed

4)      Check punctuation, missing points. 
Thanks so much, I checked the punctuation and missing points with the help of the editor this time.

Reviewer 3 Report

This paper studies the formation of bubbles during pulsed electric fields use and the resulting limits in electroporation applications. 

1) Authors change the title, focusing now in medical an electrolysis. 

These are two completely different applications with diverse specifications. Introduction should be rewritten.

2) The main question is that, electrolysis and medical applications with PEF have completely different protocols, and the bubble effect does not affect both in the same way.

The electrolysis effect is not described in a very similar studied about bubble formation in medical applications:

Samo Mahnič-Kalamiza, Damijan Miklavčič, Scratching the electrode surface: Insights into a high-voltage pulsed-field application from in vitro & in silico studies in different fluid, Electrochimica Acta, Volume 363, 2020, 137187, ISSN 0013-4686, https://doi.org/10.1016/j.electacta.2020.137187.

This should be discussed.

I believe the similarities authors make between these applications are not correct.

3) The pulse protocol used for simulation is not used in medical applications. Nowadays, ns bipolar pulses are used, not long 100s µs pulse. 

Why was the pulse protocol used in this work chosen, what is the similarities with the applications described.

4) Also, what is the effect of the electrode’s geometry in these effects. In most of the applications, planar parallel electrodes or co-linear ones are used. This affects the electric field distribution and the described effect.

Author Response

Hello dear. Thanks so much for your review again. this time answer your line by line your comments. into word document.

Round 3

Reviewer 3 Report

This article studies bubble formation during the use of pulsed electric fields for several applications

Considering the detailed answer given to reviewer last comments, it will be important to include in the Introduction section some of described justification for the used protocol.

Also, because, authors say: “[4]. Application of PEF therapy is a procedure using an intense but short electric pulse ….” In line 29. Which is not the case in this paper.

It should be included in the Introduction what are the pulse protocols that authors are referring too: e.g. 8 pulses of 100us, to give 800us. Otherwise this is not justified. Consequently, readers can be led to think about other protocols, where bipolar or 100 ns pulses are used, which is very common nowadays.

Finally, change the word “despicable” in line 137, which is not appropriate.

Author Response

Dear, thank you very much, I attach my answers in a word file.
